# An Optimization Method for Radar Anti-Jamming Strategy via Key Time Nodes

Cheng Feng [1] , Xiongjun Fu [1,2,*], Jian Dong [1], Zhichun Zhao [1], Jiyang Yu [3] and Teng Pan [3]

1 School of Integrated Circuits and Electronics, Beijing Institute of Technology, Beijing 100081, China; 3120195384@bit.edu.cn (C.F.); 7420220083@bit.edu.cn (J.D.); 3220221552@bit.edu.cn (Z.Z.)
2 Tangshan Research Institute of BIT, Tangshan 063007, China
3 Beijing Institute of Space Systems Engineering, Beijing 100094, China; yujiyang@spacechina.com (J.Y.)
* Correspondence: fuxiongjun@bit.edu.cn

**Abstract:** This paper proposes an optimization method to improve the radar anti-jamming strategy by using the predictability of left game interval. Firstly, we propose the concept of key time nodes in radar/jammer confrontation and analyze its predictability. Secondly, we analyze the radar-winning scenarios by considering the temporal constraints and construct the actual utility matrix of the radar. Then, we describe the optimization algorithm using radar-winning probability statistics based on the prediction of left game interval. Finally, we carry out a simulation experiment by comparing it with other anti-jamming strategies to verify the rationality, and the result shows that the proposed method can significantly improve the radar's winning probability in the confrontation. By using the proposed anti-jamming strategy optimization method just at the key time nodes, the imperceptibility from the jammer is improved, and its long-term superiority can be maintained in the confrontation.

**Keywords:** radar; anti-jamming; game theory

## 1. Introduction

A radar is an important sensor in missile attacks and plays a key role in electromagnetic spectrum warfare, such as in the scenario of a naval vessel assault with a missile. However, jamming technology is developing rapidly, and the electromagnetic environment on the battlefield is becoming increasingly complex, posing serious threats and challenges to the detection accuracy of the radar [1]. The confrontation between the radar and the jammer is dynamic and continuous [2]. With proper anti-jamming actions, the radar can occupy a dominant position in one confrontation round. However, the radar has to ensure its dominance in the last round to beat the opponent. When facing one jamming action, the radar may have several anti-jamming actions that can counter the jammer. It is essential to seriously consider how to choose the optimal anti-jamming action to win in the last round as well as in the current one. Therefore, the anti-jamming strategy of the radar must be optimized to complete the detection and anti-jamming task when detecting the naval vessel.

When the radar moves rapidly towards the naval vessel, the vessel keeps adjusting its location to avoid being detected easily. Therefore, the radar must detect the vessel's location constantly. However, the naval vessel uses various types of jamming to protect itself, and the radar must use anti-jamming actions accordingly to ensure the performance of its detection ability. The radar has a set of detection and anti-jamming actions which can be used to suppress the jamming actions, while the naval vessel has a set of jamming actions which can be used to jam the radar. Neither the radar nor the naval vessel has an action that can defeat all the opponent's actions, which makes the anti-jamming research meaningful. The confrontation between the radar and the naval vessel can be regarded as a two-person, zero-sum dynamic game. With the competition between jamming and anti-jamming technology becoming fiercer, there is increasing attention on radar anti-jamming

techniques from radar researchers. Game theory is a useful tool in analyzing the strategies in the radar and jammer confrontation [2].

The game theory is often applied to solve the Nash Equilibrium (NE) problem in radar anti-jamming situations. Ref. [3] proposed a game-theory-based radar anti-jamming model, which depicted the process as information collection and identification, evaluation of utility functions, and optimization of results. The model can be applied in a static game, but it is not proper in a dynamic confrontation. Ref. [4] studied the radar anti-jamming process in the context of a signal game and conducted simulation experiments but lacked the process of action recognition, which is very important in the radar anti-jamming process. Ref. [5] analyzed and summarized the scenarios of radar anti-jamming strategy based on game theory. Mutual information was considered to optimize the utility function, and equilibrium problems were analyzed in both symmetrical and asymmetrical game information. The mutual information criterion was also used to solve the strategy design problems in the framework of Stackelberg and the egalitarian games in [6]. Ref. [7] studied radar anti-jamming strategy in imperfect information scenarios and compared several methods of getting NE points. However, it did not consider that the game interval was limited and would influence the confrontation result compared to the unlimited condition. Ref. [8] studied anti-jamming methods for SAR and obtained solutions under the conditions of imperfect and perfect information, respectively. Ref. [9] analyzed the effects of jamming on the target detection performance of a radar anti-jamming method, which used constant false alarm rate (CFAR) processing and tried the resulting matrix-form games in finding the optimal strategies for both the naval vessel and radar in the game. Ref. [10] studied a distributed beamforming and resource allocation technique for a radar system in the presence of multiple targets and proved the existence of NE and its uniqueness in both partially cooperative and noncooperative games. Ref. [11] proposed a game confrontation model based on the non-real-time characteristics of radar and jammer actions, which could be used to make decisions during the confrontation. However, its optimization was achieved under the assumption that the game may end at any time. In an actual confrontation, the left game interval can be predicted by using the prior information. Ref. [12] proposed an optimization method that predicted the actions of the jammer when the radar chose the actions based on the Stackelberg game. It only considered the radar actions for one more round, which was not enough in the confrontation. Ref. [13] proposed a distributed joint allocation method of power and bandwidth based on a cooperative game for the problem of multi-resource joint allocation in a netted radar system. However, it did not consider the jammer when detecting the target. Ref. [14] explored the intelligent game between a subpulse-level frequency-agile (FA) radar and a transmit/receive time-sharing jammer under the jamming power dynamic allocation, which utilized neural fictitious self-play (NFSP) to optimize the confrontation between the radar and the jammer with imperfect information. Ref. [15] utilized an extensive-form game (EFG) with imperfect information to model the multiple rounds of interaction between the radar and the jammer. Ref. [16] proposed an adaptive weight method combined with Jensen's inequality to solve the problem of the summation of multiple weighted targets. In addition, the jammer can also use game theory to jam the radar. Ref. [17] proposed an intelligent system based on reinforcement learning, which used a virtual jamming decision-making method to enable the jammer to learn and jam efficiently without the user's prior information.

The main purpose of the above literature focuses on the confrontation benefits of the player's specific action. However, winning with a radar usually requires superiority in the last round of a game. By predicting the left game interval of each confrontation round, prior information is used to optimize the anti-jamming strategy. Meanwhile, the above literature did not consider the concealment of strategies. After the radar adopts a new anti-jamming strategy, it may gain an advantage for a period. The new strategy may be noticed by the opponent in repeated games, resulting in the same or similar strategies being taken and returning the game between the players back to a symmetrical situation [12]. Therefore, it is necessary to enhance the concealment of strategies.

To solve the above problems, this paper proposes an anti-jamming strategy optimization method via key time nodes to depict and model the dynamic confrontation between a missile-borne radar and a shipborne jammer. The proposed method improves the detection and anti-jamming capability of the radar by predicting the left game interval. Adopting the proposed optimization strategies only at key time nodes can enhance the concealment of the strategy. The main contributions of this paper are as follows.

In the game model with temporal constraints, we modeled the decision process of the jammer with the Markov Decision Process (MDP), and we built paths for virtual adversarial actions between the radar and the jammer. We predicted the left game interval by comprehensively considering the distance between the radar and the jammer, the velocity of the radar, and the planning trajectory, which were used to select the optimal radar actions of each confrontation round.

The game process was divided into regular stages and key time nodes, with the latter accounting for a relatively short time. The proposed optimization method was only used at the key time nodes, which brought great concealment of the radar strategy.

The simulation experiments contained the comparison with the method in [12], which optimizes the radar anti-jamming strategy through reinforcement learning to get the radar dominance range for as long as possible, with a regular strategy that chooses the anti-jamming action with the maximum benefit when facing the current jamming actions. The experimental result proved that the radar winning probability optimized using the proposed method was significantly improved.

The remainder of this paper is organized as follows. Section 2 presents a confrontational game model of a radar and a jammer, the concept analysis of key time nodes, and the jammer decision process model. In Section 3, the confrontation paths of the radar and jammer are built, and the radar strategy optimization method based on statistics is proposed and analyzed. Section 4 carries out simulation experiments and compares the results with the method in [12] and a regular strategy. The results demonstrate that the winning probability of radar detection and anti-jamming is greatly improved using the proposed method, verifying the effectiveness of the algorithm. The experimental results are discussed in Section 5, and the conclusions of this study are presented in Section 6.

## 2. Materials

First, we analyzed the concept of key time nodes in this section, which are short periods that play an important role in the confrontation. Then, we built the radar/jammer model with temporal constraints. Then, a round of the game was divided into four parts, including the recognition of jamming actions using the radar, the preparation of anti-jamming actions, the recognition of radar actions using the jammer, and the preparation of jamming actions. If the game ends within the interval of the former two stages, the jammer wins the confrontation, otherwise the radar wins. In [12], they increased the proportion of the radar dominance interval in a confrontation round as much as possible to win the game, as it was assumed that the game may end at any time. However, we can predict the left game interval, which can be used to optimize the anti-jamming strategy. In other words, there is a law for when the game will come to an end.

### 2.1. Concept and Analysis of Key Time Nodes

Some periods have more influence on the result than other periods, which are called key time nodes in the confrontation process between the radar and the jammer.

The duration distribution of the radar at different working stages has a significant impact on the result of the game. When a radar-guided missile attacks a naval vessel, it needs to skim the sea for quite a long flight and then quickly pull up to launch an attack when approaching the target. The duration of the latter stage is short but plays a key role. The duration of the radar in each working stage has a great impact on the final game result.

Radar operating modes also have an impact on the result of the game. The radar poses varying degrees of threat to a naval vessel in different operating modes. In search

mode, the radar beam has a wide range of illumination and pays low attention to a naval vessel, resulting in a low level of threat. In tracking mode, the radar beam continuously illuminates the naval vessel, posing a very high level of threat. Different operating modes will lead to corresponding jamming actions, which may produce different confrontation results. The duration of different radar operating modes has a great impact on the final game.

Considering the impact of factors such as work stage duration and operating modes, it can be concluded that during the confrontation process, some nodes are more important than others and have a greater impact on the outcome of the game. These nodes are called key time nodes. For example, the time nodes are very important when the radar successfully captures the target and changes from search mode to tracking mode, which can enable the predominance of the radar in the detection process, greatly improving the detection success rate. Additionally, the time nodes when the naval vessel enters the non-escape zone of the radar-guided weapon affect the game result greatly, when the radar can complete the detection task with a very high probability. There is a higher probability of completing detection and anti-jamming tasks for the radar when making favorable action choices at these key time nodes.

Additionally, by applying a novel anti-jamming strategy only at the key time nodes and a regular strategy in other adversarial periods, the concealment of the proposed strategy can be improved. This concealment can avoid the problem of the opponent learning our strategy during a long-term confrontation, extending the period in which the radar plays a leading role by proposing an optimized method. This work takes the key time nodes of when the naval vessel enters the non-escape zone of the radar-guided weapon as an example to optimize the anti-jamming strategy.

### 2.2. Game Model between the Radar and the Jammer

The elements of the game theory generally include players, strategy sets, and utility functions [18]. The players are the participants in the game, the strategy sets are the sets of actions that can be taken by each player, and the utility function is the benefit that corresponds to the actions taken by the radar and the naval vessel, respectively. The strategy set and utility functions of one player are generally unknown to the other. However, they can be estimated using prior knowledge of the confrontation between the radar and the naval vessel.

The utility matrix is the matrix of benefits that the players can get when they take different actions. It is assumed that the radar and the naval vessel both know the utility matrix of the opponent. The actions of the naval vessel include Barrage Noise (BN), Responsive Spot Noise (RSN), Doppler Noise (DN), Range False Targets (RFT), and Velocity Gate Pull-Off (VGPO) or other jamming attacks. The actions of the radar are simply regarded as Anti-BN, Anti-RSN, etc. The numbers of actions taken by the radar and naval vessel are $m$ and $k$. The utility matrix of the radar is expressed with Equation (1):

$$\mathbf{E} = \begin{pmatrix} e_{11} & e_{12} & & e_{1k} \\ e_{21} & e_{22} & \cdots & e_{2k} \\ \vdots & & \ddots & \vdots \\ e_{m1} & e_{m2} & \cdots & e_{mk} \end{pmatrix} \tag{1}$$

The rows represent the radar actions, and the columns represent the naval vessel actions. $e_{ij}$ is the benefit of the $i$th action of the radar to the $j$th action of the naval vessel, which represents the radar capability to obtain information of the naval vessel. It can be reflected with parameters such as signal-noise-ratio (SNR), the detection probability (Pd), and so on. If $e_{ij}$ is positive, the radar holds the leading position. Otherwise, the naval vessel holds the leading position. A large value of $e_{ij}$ means the radar can get the information of the target easily and accurately. According to the characteristics of the zero-sum game, the utility matrix of the naval vessel is $-\mathbf{E}$.

*2.3. Winning Condition Analysis of Radar Anti-Jamming Game*

We need to analyze the winning condition to clarify what strategy is useful in confrontation. Attention should be paid to the temporal constraints of actions in the game as the total game interval is limited in actual process, and it consumes time for the actions of the radar and naval vessel to take effect, as well as for them to be recognized by the opponent. These temporal constraints have great effects on the game result. The interval that the radar and the naval vessel need to take effect is called the preparation interval, and the interval required to recognize the opponent's action is called the recognition interval.

After the radar's action takes effect, the naval vessel carries out the recognition process and takes its action, which is then recognized by the radar. The preparation interval of the radar's action is recorded as $T_p(n, a)$, and the recognition interval of the naval vessel to the radar's action is recorded as $T_R(n, a)$. The preparation interval of the naval vessel's action is recorded as $T_P(n, s)$, and the recognition interval of the radar to the naval vessel's action is recorded as $T_R(n, s)$. These four stages are called a round of confrontation in the game.

The game is composed of multiple rounds where the number is determined by the total game interval, the recognition interval, and the preparation interval of the radar and jammer in each round.

The result of the game between the radar and the jammer is usually determined by the result of the last round, which is quite different from the general two-person, zero-sum game.

In the last round of the game, if the left game interval is less than the sum of the recognition interval and the preparation interval of the naval vessel, the radar wins the game, otherwise the naval vessel wins the game. The dominant ranges of the radar and the naval vessel are shown in Figure 1. The radar anti-jamming strategy should be optimized to make the game end at the blue ranges with more probability.

| Radar Dominant Interval Range | | Jammer Dominant Interval Range | |
|---|---|---|---|
| Jammer Recognition Interval | Jammer Action Preparation Interval | Radar Recognition Interval | Radar Action Preparation Interval |

**Figure 1.** Interval distribution of radar dominance in one round.

## 3. Methods

In the radar/jamming game, we modeled the decision process of the jammer with the Markov Decision Process, and we built paths for virtual adversarial actions of the radar and jammer. We predicted the left game interval by comprehensively considering the distance between the radar and the jammer, the velocity of the radar, and the planning trajectory, which were then used to select the optimal radar actions through the statistics of each radar action winning probability.

*3.1. Predictability of Left Game Interval*

Ref. [12] proposed a reinforcement learning optimization method by maximizing the duration ratio of radar dominance in a single round to model the temporal constraints of the game. It was assumed that the confrontation may end at any time, therefore, the optimization goal was to increase the dominant duration ratio of the radar in a single round. The radar tried to shorten the recognition interval and the preparation interval of the radar and prolong the above two of the jammer.

However, in the actual confrontation between the radar and the jammer, the radar can predict and estimate the left game interval. This is due to the radar's ability to obtain prior information such as the distance between the missile and target, radar platform velocity, and radar platform trajectory planning. By utilizing this predictability, the left game interval can be used as prior information to optimize radar anti-jamming strategies and place the radar in a dominant position at the end of the game, thereby increasing the

radar winning probability. The Optimization method diagram using the prediction of the left game interval is shown in Figure 2.

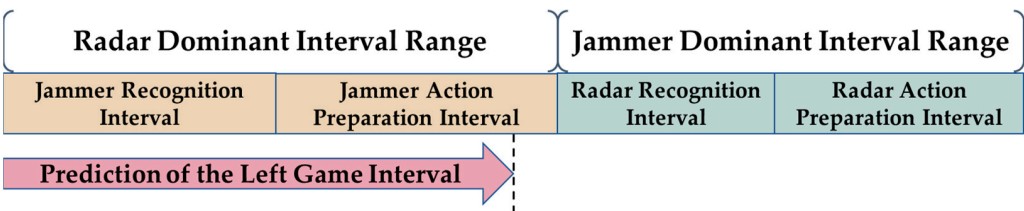

**Figure 2.** Optimization method diagram using the prediction of the left game interval.

Utilizing the predictability of the left game interval for optimization requires a high accuracy of estimation. Taking the time node of when a radar-guided missile tracks the naval vessel to ensure the latter entering the non-escape zone as an example, the radar cannot predict the left game interval accurately when the missile target is far, which may result in significant errors. During the confrontation process, the naval vessel may move to other places. Although the movement velocity is relatively low compared to that of the radar platform, it may affect the missile target distance and radar platform trajectory planning due to the long game interval, thereby affecting the prediction accuracy of the left game interval.

When the prediction accuracy of the left game interval is not high enough, such as in the condition where the prediction error is comparable to the length of one of the four stages in a round of the game, there will be significant uncertainty in the prior information. There is a high probability that the radar cannot dominate at the end of the game as expected by using the left game interval in this condition for strategy optimization, thereby invalidating radar decision-making methods based on the predictability of the left game interval.

The left game interval should not be too long to ensure a high prediction accuracy. In the game between the radar and the jammer, the predicted value of the left game interval can be continuously updated in each round. As the game goes on, the accuracy of predicting the left game interval will become increasingly accurate. During several rounds just before the game ends, the radar can achieve high accuracy in estimating the left game interval, meeting the conditions for optimizing the radar anti-jamming strategy.

The proposed optimization method sets a left game interval initialized to a positive value, which is short enough to be estimated precisely. At the end of each round of the radar/jammer game, an updated value can be obtained by subtracting the game interval of the current round from the left game interval. Repeat this process until the left game interval decreases to 0 (or from a positive value to a negative one) and ends the game.

### 3.2. A Markov-Decision-Process-Based Jammer Model

In the radar/jamming game process, both players continuously make action choices. We constructed a decision-making process model for the actions of the jammer through the Markov Decision Process [19] and optimized the anti-jamming strategy through the interactions with this jammer model.

The MDP lacks aftereffect, as the current state is only related to its previous state and is not related to other earlier states. Since the probability of state transition at a certain moment only depends on its previous state, after the transition probability between any two states in the model is calculated, we can determine the model of MDP.

For any $n \geq 0$ and state $i, j, i_0, i_1, \cdots, i_{n-1}$, there is:

$$P\{X_{n+1} = j | X_0 = i_0, X_1 = i_1, \cdots, X_{n-1} = i_{n-1}, X_n = i\} = P\{X_{n+1} = j | X_n = i\} \quad (2)$$

where $P\{X_{n+1} = j | X_0 = i_0, X_1 = i_1, \cdots, X_{n-1} = i_{n-1}, X_n = i\}$ is the probability that the next state of $X$ comes to the state $j$ under the condition of $X_0 = i_0, X_1 = i_1, \cdots, X_{n-1} = i_{n-1}, X_n = i$, which is the same as under the condition of $X_n = i$. Once the

initial distribution of the MDP $P\{X_0 = i_0\}$ and the mapping $P(a_t|S_t = s_t)$ of state $S_t$ to return $a_t$ are determined, its statistical characteristics are completely determined by the conditional transition probability $P\{X_{n+1} = j|X_n = i\}$.

This section constructs a conditional transition model for the jammer state through MDP. The jammer adopts a new jamming action based on the conditional transition probability for the current radar action. The conditional transition probability of jammer action can be described with the softmax rule and calculated according to Equation (3):

$$P(\text{i}) = \frac{e^{-u_i}}{\sum\limits_{j=1}^{n} e^{-u_j}} \tag{3}$$

where $P(\text{i})$ represents the probability of choosing the $i$th jamming action from the $k$ jamming candidates when facing the radar cation. It means that when the jammer faces the actions taken by the radar, the probability of selecting the jamming action with the largest utility is the highest, while other jamming actions could also be selected.

In the regular decision-making process of the jammer, the transfer of jammer actions only considers the radar action that the jammer is currently facing, while the influence of other factors is relatively small and ignorable. It is assumed that the basis for selecting jammer actions is only the conditional transition probability.

Figure 3 shows the process of the jammer taking actions based on the probability of state transition. Among the states, $S_1$~$S_4$ are the actions available for each round, and their transfer probability can be calculated by the benefits between different actions in round $t$ and round $t + 1$. For example, when the state $t = 0$ turns to $t = 1$, the transfer benefits of the four actions $S_1$~$S_4$ from the initial action $A$ to round $t = 1$ are $a_{01}$~$a_{04}$, respectively. The probability of action transfer to $S_1$~$S_4$ can be calculated using Equation (3), and the next action can be selected according to the probability distribution. Afterwards, the left game interval is updated and processed in the same way till the end. A complete path can be obtained, as shown $A \rightarrow S_4 \rightarrow S_2 \rightarrow S_3 \rightarrow S_3$ in Figure 3.

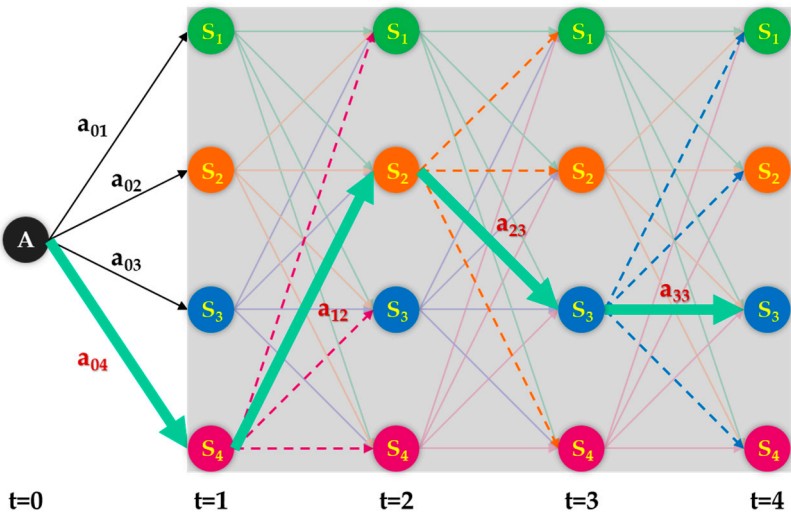

**Figure 3.** Schematic diagram of a jammer action transfer based on conditional transfer probability.

A jammer decision-making model based on MDP is constructed through the above description. The jammer selects its jamming actions using the softmax rule when facing current radar actions. Then, the radar can optimize its strategy through the interaction with the jammer model, forming a game model of "jamming/anti-jamming/jamming/anti-jamming".

### 3.3. A Radar Action Selection Method Based on Statistics

When the radar faces the current jamming action, it has to select one of multiple actions in the strategy sets to suppress the jamming, then the jammer needs to select a jamming action based on MDP. Both players in the game repeat the process till the game ends. A path of confrontation is then formed. The process of building the confrontation paths is shown in Figure 4.

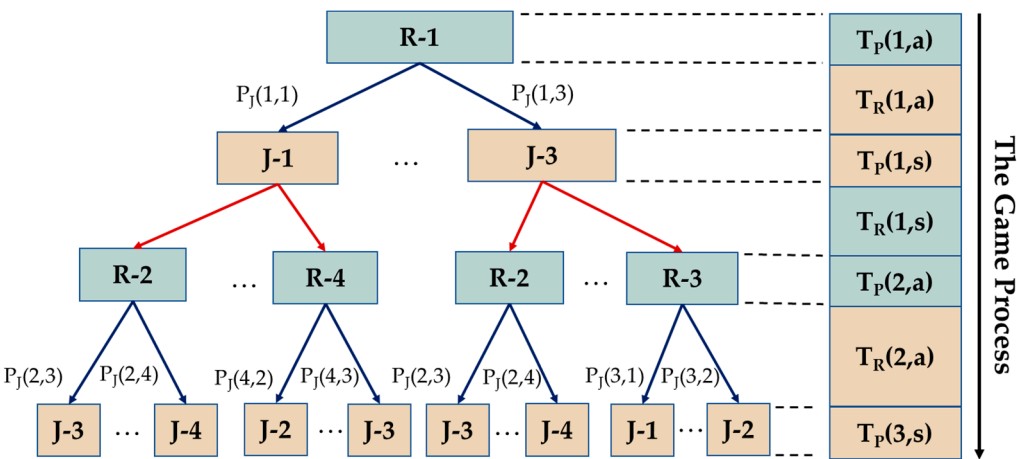

**Figure 4.** Schematic diagram of radar/jammer action confrontation paths.

There are several options when one player makes the choice. All the possible actions of the radar and jammer form a 2n-layer N-ary tree [20], in which $n$ is the confrontation round and $N$ is the number of paths they build in confrontation. Among them, the blue box represents the actions of the radar, and the brown box represents the actions of the jammer. The blue line from the blue box to the brown box represents the possible transfer process of the jammer, and the content marked on the line is the transfer probability. For example, $P_J(1,1)$ is the probability of the jammer choosing the first jamming action when facing the first type of radar action. The red line with a brown box pointing towards the blue box does not indicate the transfer probability, which is determined with the radar anti-jamming strategy. The content marked on the right represents the time consumed by each game stage. The preparation interval of the radar action is $T_p(n,a)$, the recognition interval of the jammer for the radar action is $T_R(n,a)$, the preparation interval of the jammer action is $T_P(n,s)$, and the recognition interval of the radar for the jammer action is $T_R(n,s)$. In Figure 3, $a$ is the radar action, and $s$ is the jammer action.

Among those paths, the radar needs to select the one that can bring the highest winning probability. When making a choice, the radar selects its action through the statistics of virtual winning probability. The flowchart of selecting the radar actions is shown in Figure 5.

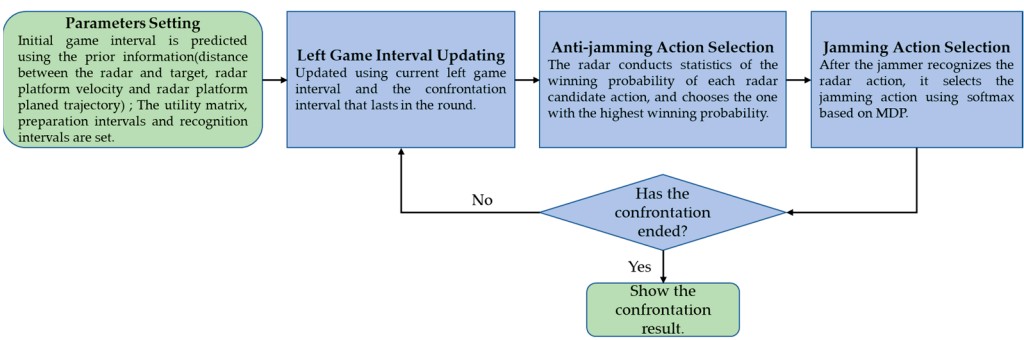

**Figure 5.** Flowchart of the optimization method via key time nodes based on statistics.

The optimization algorithm of selecting the radar action is as follows (Algorithm 1):

---

**Algorithm 1:** Selection method of radar anti-jamming actions based on statistics.

---

**Step 1:** Set the left game interval to $T_0$, the left game interval copy to $T_1$, and let the assignment $T_0 = T_1$.

Set the jammer to select the current action with a probability starting at 1. The initial number of the radar candidate action $i$ is 1. The initial number of the jammer candidate action $j$ is 1.

**Step 2:** The current action of the jammer is identified and labeled as $J_C$.

**Step 3:** According to the utility matrix, the action of $J_C$ with positive utility for this jamming action is selected as the candidates of the radar. Obtain the total number $N$ of radar candidate actions and record the number of each radar candidate action to form a sequence $R(1) \cdots R(N)$.

**Step 4:** Select the radar candidate action with serial number $R(i)$, assuming it is selected as the next radar action.

**Step 5:** According to the utility matrix, the jammer selects the action with negative utility for this radar action $R(i)$ as the candidate of the jammer. Obtain the number of jammer candidate actions M and record the number of each jammer candidate actions to form a sequence $J(1) \cdots J(M)$.

Calculate the probability of moving to each jamming action according to the softmax rule. Select the action of jammer serial number $J(j)$ and record the probability of switching to this jamming action $P_1(J(j))$:

$$P_1(J(j)) = \frac{e^{-u(R(i),J(j))}}{\sum_{j\_loop=1}^{M} e^{-u(R(i),J(j\_loop))}} \tag{4}$$

**Step 6:** Update the probability that the jammer selects the current action $P_0$:

$$P_0 = P_0 * P_1(J(j)) \tag{5}$$

**Step 7:** Update the left game interval $T_0$ according to Equation (6) and determine whether its value is positive:

$$T_0 = T_0 - T_R(J_C) - T_P(R(i)) - T_R(R(i)) - T_P(J(j)) \tag{6}$$

If the result is negative, the game is judged to finish.

Continue to determine whether the radar wins at the end of the game.

If the radar wins, the probability of the radar choosing the current action path is recorded, and the winning probability of the radar under $N(i)$ node is updated:

$$P_R(R(i)) = P_R(R(i)) + P_0 \tag{7}$$

Judge the relationship between $j$ and $M$, and if $j$ is not equal to $M$, update the game interval:

$$T_0 = T_0 + T_R(R(i)) + T_P(J(j)) \tag{8}$$

Let $j = j + 1$, return to **Step 5**.

If $j$ is equal to $M$, then all jammer actions have been predicted and continue to judge whether $i$ is equal to $N$. If $i$ is not equal to $N$, update left game interval according to Equation (9):

$$T_0 = T_0 + T_R(J_c) + T_P(R(i)) + T_R(R(i)) + T_P(J(j)) \tag{9}$$

Let $i = i + 1$, return to **Step 3**.

If $i$ is equal to $N$, all radar actions have been predicted, select the $R(optimal)$ actions with the largest utility from $P_R(R(optimal))$ as the next action of the radar. The jammer selects a new jamming action *optimal_new* according to the $R(optimal)$ action and its corresponding $R(i)$ action. Update $T_1$ as follows:

$$T_1 = T_1 - T_R(J_c) - T_P(R(optimal)) - T_R(R(optimal)) - T_P(J(optimal\_new)) \tag{10}$$

Return to **Step 1**.

If the result is positive, the current path is not finished. Return to **Step 2** to continue exploring.

---

In the algorithm, parameter $T_0$ represents the virtual left game interval, while parameter $T_1$ represents the real left game interval. The former is used to select the optimal radar action from the candidates, which is constant in different paths and is thereby virtual. The latter is used to update the confrontation process which will change with the game and is thereby real.

We conducted the statistics of the winning probability for every radar candidate action when facing the jamming and making choice. For each candidate radar action, we calculated the sum of the winning probabilities of all the paths with it as the vertex, in which the radar wins the game in the end. After obtaining the winning probability of all candidate radar actions, we chose the one with the highest winning probability, then moved to the next confrontation round until the left game interval turned negative.

In the selection process of the radar anti-jamming actions based on statistics, the radar selects its actions that can generate positive returns as radar candidate actions when facing the jamming actions of the current round. For each radar candidate action, their action numbers form a sequence $\{R(1), R(2), \cdots, R(N)\}$. For example, for Jamming Action 4, if Radar Action 1 and Radar Action 4 are positive for their returns, then their numbers will be formed into sequences $\{R(1), R(2)\}$, i.e., {1,4}, for subsequent winning probability calculations. When the jammer faces the radar actions of the current round, each jamming action that causes the jammer to obtain positive benefits are counted as a candidate action for the jammer and composed sequences $\{J(1), J(2), \cdots, J(M)\}$.

This process can be described according to the MDP. The alternating actions between the radar and the jammer constitute game paths and form an N-ary tree. When the radar needs to make a choice, the winning probability of each radar candidate action with it as the vertex of the confrontation path is counted. The selection probability of each radar candidate in one confrontation path is the product of the probabilities of jamming being selected at each node, as shown in Equation (5). The probabilities corresponding to the winning path results are accumulated to obtain the final winning probability of the candidate action.

The game result, which is obtained in four cases when the game ends, is shown in Figure 6.

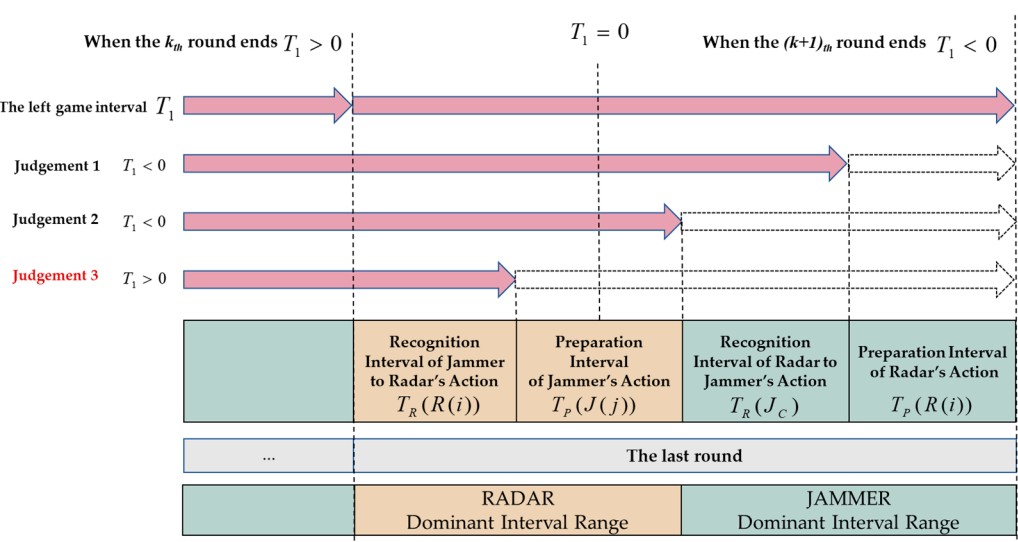

**Figure 6.** A typical judgement of the game result.

We divided the result of the confrontation into four cases.

Case 1:

$$T_1 + T_P(R(i)) > 0 \tag{11}$$

which leads to the victory of the jammer.

Case 2:

$$T_1 + T_P(R(i)) < 0 \tag{12}$$

and

$$T_1 + T_P(R(i)) + T_R(J_C) > 0 \tag{13}$$

which leads to the victory of the jammer.

Case 3:

$$T_1 + T_P(R(i)) + T_R(J_C) < 0 \tag{14}$$

and

$$T_1 + T_P(R(i)) + T_R(J_C) + T_P(J(j)) > 0 \tag{15}$$

which leads to the victory of the radar.

Case 4:

$$T_1 + T_P(R(i)) + T_R(J_C) + T_P(J(j)) < 0 \tag{16}$$

and

$$T_1 + T_P(R(i)) + T_R(J_C) + T_P(J(j)) + T_R(R(i)) > 0 \tag{17}$$

which leads to the victory of the radar.

In Figure 4, at the end of the $k$th round of the game, $T_1 > 0$; at the end of the $(k+1)_{th}$ round of the game, $T_1 < 0$. When the round of the game is determined to be the last one, the judgement of whether the radar is in an advantageous position starts. By analyzing the above four stages in sequence, it can be found that in the second judgment $T_1 + T_P(R(i)) + T_R(J_C) < 0$, in the third judgment $T_1 + T_P(R(i)) + T_R(J_C) + T_P(J(j)) > 0$, and it belongs to Case 3, so it is determined that the radar is in an advantage.

### 3.4. Algorithm Computation Overload Analysis

The computational complexity of the proposed algorithm mainly lies in the statistics of the victory of the radar candidate actions in each confrontation path. In each round of the game, assuming $n$ rounds of games are played before the key time nodes, with $M$ candidate radar actions in each round. For each radar action, the jammer can have $N$ actions to choose from.

In one round of the game, a total of $M * N$ paths are formed; for $n$ rounds of games, a total of $(M * N)^n$ paths are formed.

The interval required for the radar to recognize the jamming action, the radar action preparation, the jammer's recognition of the radar action, and the jammer action preparation varies in a single round of game on each path. However, due to the limited action choices of the radar and the jammer, specific values can be obtained by looking up tables during a single round interval calculation. By storing data in advance, the addition operations of a single round are 3, therefore $3n * (M * N)^n$ addition operations are required.

For each path, it is necessary to calculate the radar winning probability, which is obtained by multiplying the conditional transition probability of the jammer in each round of the game, and $n * (M * N)^n$ multiplication operations are required. The conditional transition probability of the jammer can also be stored in advance, and specific values can be obtained by looking up tables without the need for multiplication and addition statistics.

At the end of each path, it is necessary to calculate whether the radar is in an advantageous position, as shown in Figure 4. At most four, and at least two, comparisons are required, and three are taken as a typical value. The number of comparisons is then $3n * (M * N)^n$, which means $3n * (M * N)^n$ addition operations are required.

Therefore, for $n$ rounds of games, $n * (M * N)^n$ times multiplication and $6n * (M * N)^n$ times addition operations are required.

From the above analysis, it can be seen that as the number of game rounds increase, the multiplication and addition operations required by the algorithm will sharply increase, and the more action choices of radar and jammer, the faster the computational overload required by the algorithm.

However, as mentioned ahead, to ensure the effectiveness of the algorithm, the dynamic prediction accuracy of the key time nodes must be high enough. It requires that the number of game rounds should not be too large, which will limit the computational complexity of the algorithm.

In addition, in the discussion of $n$th rounds of games, all possible paths of the game have been included. After the jammer selects an action and enters the $(n+1)$th round of the game, there is no need to recalculate. We can search for the calculation results of the previous corresponding sub node N-ary tree and use the conditional transition probability of the corresponding jammer action to correct the winning probability of this round. The algorithm process is similar to dynamic programming [21], but it cannot be recursive. Instead, it can be used by subsequent nodes to query and call after completing

the calculation at one time, and the desired results can be obtained after minor adjustments. Adopting this strategy of exchanging space for time can improve the operating speed of the algorithm.

## 4. Results

This section conducts simulation experiments to verify the effectiveness of the algorithm. The proposed radar anti-jamming strategy optimization method via key time nodes of the game will be compared with the strategy optimization method of [12] on radar/jamming actions, as well as the regular strategy for selecting the radar action with the highest utility for current jamming. The experiment data are the same with the strategy optimization method of [12].

### 4.1. Experimental Settings

When applying the optimization algorithm for the radar anti-jamming strategy via key time nodes proposed in this work, the following parameters should be set based on prior knowledge: the utility matrix, the radar action preparation interval and recognition interval to the jammer action, and the jamming action preparation interval and recognition interval to the radar action.

According to Section 2.2, the radar anti-jamming utility matrix in the simulation is as follows:

$$\mathbf{E} = \begin{pmatrix} +2 & -3 & +1 & -4 \\ -2 & +3 & -4 & +1 \\ -3 & +2 & +4 & -3 \\ +1 & -5 & -3 & +3 \end{pmatrix}$$

The rows represent the radar actions, and the columns represent the naval vessel actions. $e_{ij}$ is the benefit of the $i$th action of radar to the $j$th action of naval vessel, which represents the radar capability to obtain information of the naval vessel. It can be reflected using parameters such as signal-noise-ratio (SNR), the detection probability (Pd), and so on. If $e_{ij}$ is positive, the radar holds the leading position. Otherwise, the naval vessel holds the leading position. A large value of $e_{ij}$ means the radar can get the information of the target easily and accurately.

The three algorithms in this simulation verification use the same parameters to highlight the effectiveness of the proposed algorithm. The actions of jammers include barrage noise (BN), reactive spot noise (RSN), doppler noise (DN), range false targets (RFT), and velocity gate pull off (VGPO) or other jamming attacks. The actions of the radar are simply regarded as Anti-BN, Anti-RSN, etc. Choosing four jamming actions and four radar actions, we determined the utility matrix of radar/jamming based on prior knowledge, as well as their respective preparation and recognition intervals for the experiment.

The radar preparation interval and recognition interval to the jammer action are set in Table 1, and the jamming action preparation interval and recognition interval to the radar action are set in Table 2.

**Table 1.** Preparation and recognition intervals of the radar.

| Radar Action Number | Preparation Interval $T_P(a)$ (0.01 s) | Recognition Interval of Jammer to Radar $T_R(a)$ (0.01 s) |
|---|---|---|
| 1 | 5 | 4 |
| 2 | 2 | 1 |
| 3 | 2 | 2 |
| 4 | 3 | 3 |

**Table 2.** Preparation and recognition intervals of the jammer.

| Jammer Action Number | Preparation Interval $T_P(s)$ (0.01 s) | Recognition Interval of Radar to Jammer $T_R(s)$ (0.01 s) |
|---|---|---|
| 1 | 6 | 1 |
| 2 | 4 | 3 |
| 3 | 2 | 3 |
| 4 | 1 | 5 |

The typical left game interval for the simulation was set as 0.72 s, within which the radar and jammer could engage in 6–8 rounds of confrontation. In Section 3.1, the application scenarios of this proposed method are discussed. If the left game interval was set too long, it would lead to significant errors in the prediction results. If the error could be compared to the duration of one of the game stages, the prediction error could be too large, and the optimization method becomes ineffective. At the same time, it would cause the number of branches in the N-ary tree to increase exponentially, causing a rapid increase in computational complexity and reducing the real-time performance of policy calculations.

*4.2. Experimental Result*

In the simulation experiment, the proposed algorithm used the left game interval of dynamic prediction as prior information to optimize the radar anti-jamming strategy. After the radar made the decision, the probability of winning the game in this round was predicted, and then the radar winning probability in this round was updated after the radar/jamming confrontation entered the next round. The radar winning probability in each game round could be obtained by repeating this process continuously. This method, the Q-learning optimization method in [12], and a regular strategy for radar/jammer confrontation, which always chose the radar action with the largest utility when facing the current jamming, are shown in Figure 7.

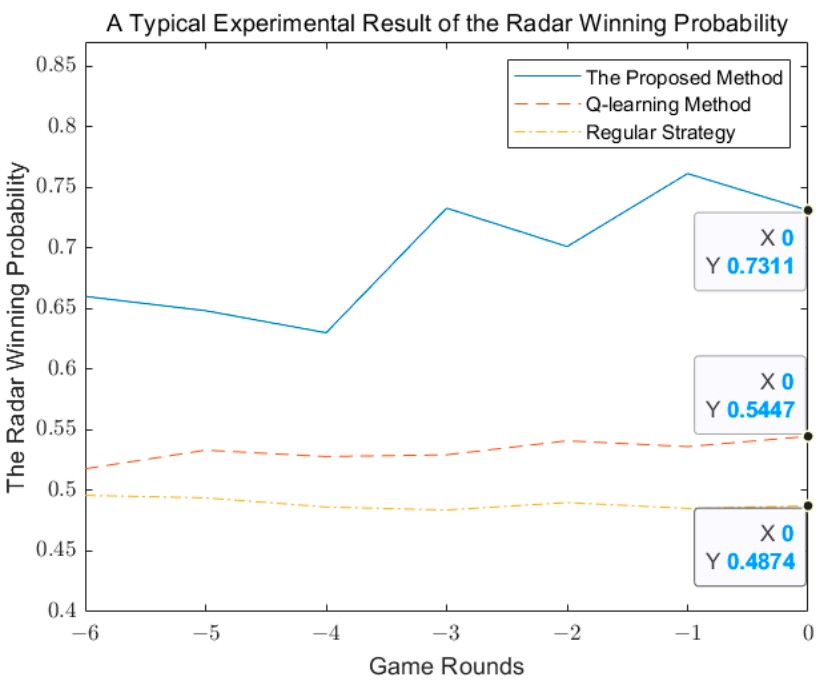

**Figure 7.** Comparison of the proposed method with other anti-jamming strategies.

In Figure 7, 0 in the horizontal axis represents the last turn of confrontation, −1 represents the game round just before the end of the confrontation, and −6 represents the sixth to the end of the game round. The proposed algorithm in this article predicted a winning probability of 73.1% when taking corresponding actions before the game ended. The Q-learning optimization algorithm proposed in [12] predicted a winning probability of 54.5%, and the winning probability of regular strategy was 48.7%.

To increase the reliability of the experiment, multiple repeated experiments were conducted under the same experimental conditions, and the simulated radar winning probability in each experiment was compared with the predicted one in a single experiment. We conducted 100 repeated comparative experiments and only calculated whether the radar won at the end of the game. To reflect the randomness in the repeated experiments, an additional random value was added to the initial left game interval $T_1$, making the simulation process more consistent with the actual scenarios.

In Figure 8, the red dots and the blue ones represent the victory and failure of the radar at the end of each experiment, respectively. The coordinates on the right represent the cumulative number of victories of the radar in 100 experiments. In 100 repeated comparative experiments, the radar won 72 times, which is basically consistent with the predicted winning probability of 73.1% of the previous radar anti-jamming experiment.

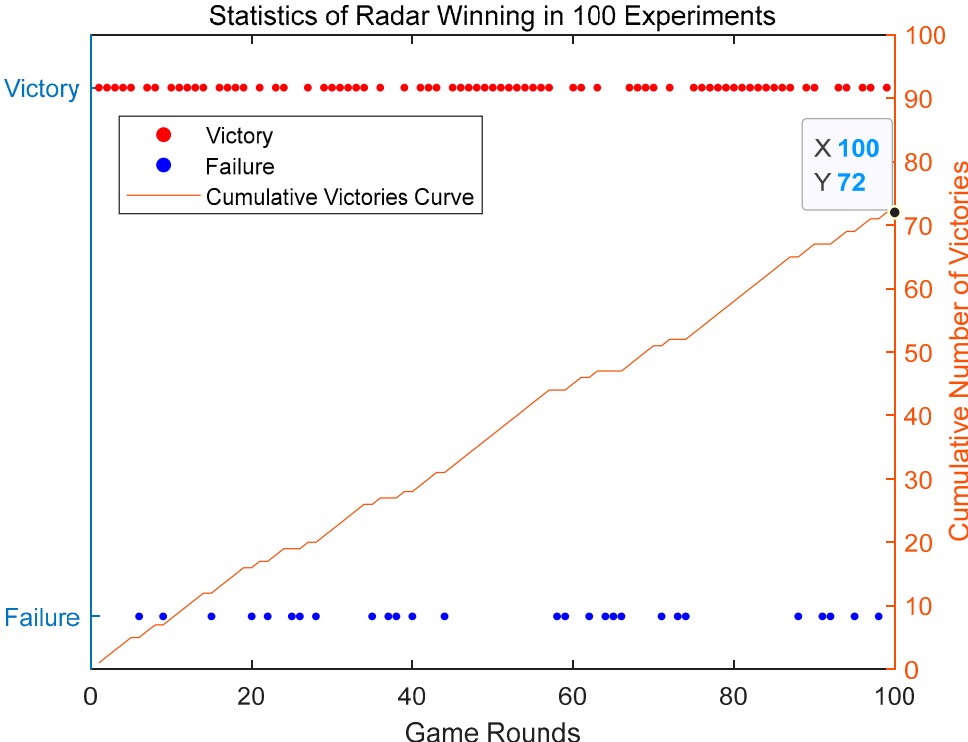

**Figure 8.** Statistical graph of the winning probability of 100 repeated experiments.

## 5. Discussion

In Figure 5, the winning probability of the regular strategy in each round of the game is stable at around 50%, and there is almost no significant change with the increase in the round of games. The winning probability of the Q-learning optimization method [12] shows a slight upward trend with the increase in game rounds, with an improvement of nearly 5% compared to a regular strategy, maintaining the winning probability at over 50%. However, the improvement effect is not significant, which is consistent with the experimental results in [12]. The proposed algorithm has a level of nearly 20% improvement in winning probability compared to Q-learning optimization methods and regular strategies, with a stable winning rate of over 60%. As the game progresses, it will gradually increase, and the predicted winning probability at the end of the game can reach over 70%. This algorithm

optimizes the decision-making process of radar actions by selecting the candidate radar action with the highest winning probability in each round.

It should be pointed out that due to the randomness of the softmax rule used for predicting jamming actions, the left game interval $T_1$ is variational, thus the judgment of the final game outcome is affected. Therefore, the winning probability of one round may not necessarily be higher than the previous round, which is also the reason why the winning probability of the proposed algorithm may decrease with the game rounds.

In Figure 6, it can be seen that the proportion of radar failure results in 100 experiments is relatively uniform, proving that multiple repeated experiments can indeed demonstrate the high accuracy of the proposed algorithm in predicting the victory probability at the end of the game. The proposed algorithm has a high randomness in determining the victory situation, and each simulation result is independent without any impact.

The correctness and effectiveness of the proposed algorithm were verified through multiple repeated simulation experiments with Q-learning optimization methods and regular strategies.

The proposed method acquires better performance as it uses prior information. Some assumptions in previous research were not consistent with the actual scenarios. We analyzed the confrontation of the radar and the jammer with temporal constraints and discovered new features that could be used for anti-jamming strategy optimization.

When the left game interval becomes larger and the confrontation rounds using the proposed method accumulate, the performance becomes more difficult to predict. As mentioned above, the left game interval will obtain more errors. Additionally, the jammer model uses the softmax repeatedly to decide its actions in each confrontation round. The softmax rule contains uncertainty, which may affect the confrontation result as the jammer may take an action that is not expected by the radar. If the jammer takes an unexpected action just in the last round, the radar has no space and limited time to optimize its action, and the confrontation result will be the same as the control group with the regular strategy. However, the probability will be very low. And, if we use this optimization many times, the average result will be much higher than the control group.

The failure of recognition to the opponent's actions is not considered in this work. In the actual confrontation, the radar or the jammer may not always recognize the opponent's actions. After an inaccurate recognition, the proposed method may be invalid as the basis of statistics becomes wrong, which is a problem that all strategies will face in the confrontation. We can continue to optimize the radar actions from a disadvantageous position after obtaining accurate recognition.

## 6. Conclusions

This work proposes a statistical-based optimization method for radar detection anti-jamming strategy, which utilizes the characteristics of key time nodes in games. When the radar needs to make decisions at key time nodes of the game, it dynamically estimates the left game interval under the MDP, calculates the sum of the winning probability of each radar candidate action, and selects the radar action based on the winning probability statistical results, which can increase the radar winning probability to over 70%. Compared to the optimization method of the radar anti-jamming strategy with temporal constraints and the regular strategy decision-making method, the winning probability was significantly improved.

This optimization strategy only targets key time nodes in the game, and the regular strategy or other game strategies can be executed at other time nodes, which can significantly increase the effectiveness of the radar strategy. This can make it difficult for the jammer to detect the new strategy adopted by the radar, thus the long-term effectiveness of this strategy optimization method can be maintained.

**Author Contributions:** Conceptualization, C.F.; Methodology, C.F. and J.D.; Software, Z.Z.; Validation, J.Y.; Formal analysis, Z.Z. and T.P.; Investigation, J.D.; Resources, X.F., J.D. and J.Y.; Writing—original draft, C.F.; Visualization, C.F.; Supervision, X.F. and T.P.; Project administration, X.F.; Funding acquisition, X.F. All authors have read and agreed to the published version of the manuscript.

**Funding:** This research was funded by 111 Project of China, grant number B14010.

**Data Availability Statement:** Not applicable.

**Conflicts of Interest:** The authors declare no conflict of interest.

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
