# Peer review of "An Optimization Method for Radar Anti-Jamming Strategy via Key Time Nodes"

_remotesensing, doi:10.3390/rs15153716_

Round 1

Reviewer 1 Report

This is a good start to discussing the optimization  of radar anti-jamming strategy. I think it has a high novelty of the work.  I just wonder the result that how to convince the real scene by the data for simulation. 

Author Response

Thanks for your appreciation! We have considered several times of how to convince the real scene by the data for simulation.

The simulation is meant to show the verification of the proposed optimization method for radar anti-jamming strategy, which must use many types of jamming and anti-jamming actions. However, collecting all the data we need is almost infeasible. Therefore, it is reasonable to use simulation to prove the effectiveness of the method rather than field experiment. As illustrated in the manuscript, we use typical parameters as experimental settings to increase the effectiveness of the experiment.

The simulation in the manuscript can show just one typical result of the confrontation between the radar and jammer. In fact, if the left game interval is predicted precisely, and the preparation intervals and the recognition intervals among different actions between the radar and jammer are distributed scattered in the confrontation, the proposed optimization method can work to increase the radar winning probability. Therefore, we believe that it can convince the real scene by the data for simulation this way.

Reviewer 2 Report

Original Submission

Recommendation

Major (very) Revision

Comments to Author:

Title:  

An Optimization Method for Radar Anti-Jamming Strategy via Key Time Nodes

Overview and general recommendation.

This paper is demonstrating usage of RADAR methods and some important applications. This technique is very important and somehow accurate and has lots of important applications. First of all, as a person with more than 20 years’ familiarity with radar/SAR, I like this paper very much; but as a scientist, I have to say the truth about the material and be honest.

The Abstract is the worst I have ever seen: It is more like an introduction. Summarize the idea and concepts. English is NOT fair; give it to a native (if you are not a one) to review (it is a big must!!). Write Abstract again!

Introduction is OK. You should follow the path that leads the readers to the central point of what you are going to present. I think in some positions, some important references (lots of new works) are missing: please fix them! Pls improve the quality of Figs> they are very very bad. Results, discussions, and conclusions are fair.

I think this paper had done good experiments. I like that. BUT, it is written very bad: it designs very bad; despite good experiments have be done; just a lot of works out and doing the corrections will do the JOB. If you do not improve the points here, my judgment is rejection!

I also think this paper does not have much to say. The material is very low, and I think this paper is not in RS MDPI level. I propose (max Impact Factor ~ 3) for this paper. Also Sensors is much better for this paper (Hardly!).

Detailed comments:

Line 9: RADAR is just not a Sensor; it is also an EM wavelength.

Eq.1: Show it in non-italic bold. It is a Matrix!

English is NOT fair; give it to a native (if you are not a one) to review (it is a big must!!).

Author Response

This paper is demonstrating usage of RADAR methods and some important applications. This technique is very important and somehow accurate and has lots of important applications. First of all, as a person with more than 20 years’ familiarity with radar/SAR, I like this paper very much; but as a scientist, I have to say the truth about the material and be honest.

  1. The Abstract is the worst I have ever seen: It is more like an introduction. Summarize the idea and concepts. Write Abstract again!

Thank you for your critics. We have summarized the idea and concepts and write the abstract again. The abstract is now as follows:

This paper proposes an optimization method to improve the radar anti-jamming strategy by using the predictability of left game interval. Firstly, we proposed the concept of key time nodes in radar/jammer confrontation and analyzed its predictability. Secondly, we analyzed the radar winning scenarios by considering the temporal constraints of non-real-time radar recognition and preparation actions and constructed the actual utility matrix of radar. Then, we described the optimization algorithm using radar winning probability statistics based on the prediction of left game interval. Finally, we carried out a simulation experiment by comparing with other an-ti-jamming strategy to verify the rationality, and the result showed that the proposed method can significantly improve the radar's winning probability in the confrontation. By using the proposed anti-jamming strategy optimization method just at the key time nodes, the imperceptibility from the jammer is improved and the long-term superiority can be maintained in the confrontation.

  1. Introduction is OK. You should follow the path that leads the readers to the central point of what you are going to present. I think in some positions, some important references (lots of new works) are missing: please fix them! Pls improve the quality of Figs> they are very very bad. Results, discussions, and conclusions are fair.

Thank you for your comments. We have added more reference in introduction. They are reference 14-18, which are fairly new.

  1. Xiaofei, P. Yu, J. Biao and Z. Zhenkai, "Joint allocation of power and bandwidth for cognitive tracking netted radar," 2021 International Conference on Control, Automation and Information Sciences (ICCAIS), Xi'an, China, 2021, pp. 263-266, doi: 10.1109/ICCAIS52680.2021.9624621.
  2. Geng, J.; Jiu, B.; Li, K.; Zhao, Y.; Liu, H.; Li, H. Radar and Jammer Intelligent Game under Jamming Power Dynamic Allocation. Remote Sens.2023, 15, 581. https://doi.org/10.3390/rs15030581
  3. Li, B. Jiu, W. Pu, H. Liu and X. Peng, "Neural Fictitious Self-Play for Radar Antijamming Dynamic Game With Imperfect Information," in IEEE Transactions on Aerospace and Electronic Systems, vol. 58, no. 6, pp. 5533-5547, Dec. 2022, doi: 10.1109/TAES.2022.3175186.
  4. Xin, F., Wang, Y., Sun, J. et al. Adaptable waveform design for radar and jammer for multi-target using game theoretic strategies. EURASIP J. Adv. Signal Process. 2022, 99 (2022). https://doi.org/10.1186/s13634-022-00932-w
  5. Zhang, S., Tian, H., Chen, X., Du, Z., Huang, L., Gong, Y. and Xu, Y. (2020), Design and implementation of reinforcement learning-based intelligent jamming system. IET Commun., 14: 3231-3238. https://doi.org/10.1049/iet-com.2020.0410

Besides, we improved the quality of the figures in this manuscript. 

  1. I think this paper had done good experiments. I like that. BUT, it is written very bad: it designs very bad; despite good experiments have be done; just a lot of works out and doing the corrections will do the JOB. If you do not improve the points here, my judgment is rejection!

We redesigned the experiment, and now it can be more easily understood and can prove the effectiveness of our proposed method more sufficiently.

  1. I also think this paper does not have much to say. The material is very low, and I think this paper is not in RS MDPI level. I propose (max Impact Factor ~ 3) for this paper. Also Sensors is much better for this paper (Hardly!).

We have reviewed carefully to improve the level of this manuscript and we hope that it can this manuscript can receive your approval. Thank you!

  1. Detailed comments:

Line 9: RADAR is just not a Sensor; it is also an EM wavelength.

We have modified our description and clarify the relevant concepts.

Eq.1: Show it in non-italic bold. It is a Matrix!

This has been corrected and thanks for your critics.

  1. English is NOT fair; give it to a native (if you are not a one) to review (it is a big must!!).

We give this manuscript to an expert in our college who received her master's degree in UK to review and we believe it is much easier to read and understand. Thanks for your critics!

Reviewer 3 Report

The author proposed the concept of key time node in radar/jammer confrontation and analyzed the predictability of the left game interval. Also, they have analyzed the radar winning conditions by considering the temporal constraints of non-real-time radar recognition and preparation actions and constructing the radar's actual utility matrix. And the same has been shown through optimization algorithms and corresponding simulation results. Overall the work is fine and redable. I have few minor suggestions

1) Proofread your paper for english 

2. Please compare the results with similar existing in the literature.

3) How you optimized the hyper-parameters of the proposed algorithm

4) How the path is decided?

5) On what basis the time interval is decided and optimized?

Minor editing of English language required

Author Response

The author proposed the concept of key time node in radar/jammer confrontation and analyzed the predictability of the left game interval. Also, they have analyzed the radar winning conditions by considering the temporal constraints of non-real-time radar recognition and preparation actions and constructing the radar's actual utility matrix. And the same has been shown through optimization algorithms and corresponding simulation results. Overall the work is fine and readable. I have few minor suggestions

1) Proofread your paper for English

Thank you for your critics. We give this manuscript to an expert in our college who received her master's degree in UK to review and we believe it is much easier to read and understand now.

  1. Please compare the results with similar existing in the literature.

The results of the work presented in the article have been compared with reference [12] and regular strategy which chooses the actions with the largest utility when facing the radar actions, which is shown in Figure 6 and Figure 7. The comparison shows the proposed method performs the best in the radar/jammer confrontation.

We have analyzed many literatures to illustrate the status of research on anti-jamming strategies, but these literatures mainly focus on the research of anti-jamming methods rather than anti-jamming strategies. Meanwhile, in the field of anti-jamming strategy optimization, there has not been a standard dataset which can be used to show the advantages and disadvantages.  In those articles about radar anti-jamming strategy, they chose Pd (Detection Probability), SNR (Signal Noise Ratio) or SIR (Signal Interference Ratio) as the optimization index, which are quite different from this work. In fact, this work extracted features from those jamming and anti-jamming actions instead of conducting radar signal processing directly. We did the research to optimize the radar decision-making strategy, rather than overcoming current specific jamming.

3) How you optimized the hyper-parameters of the proposed algorithm

There are no hyper-parameters in the proposed algorithm. The statistics of radar winning probability is the sum of probability in all winning paths, and the probability in all winning paths is the product of the jammer action choosing probability in each floor, which is decided by the softmax. Therefore, no hyper-parameters are used in the proposed algorithm.

The jammer action choosing probability in each floor is decided by softmax, which can be optimized. However, it is only used for qualitative description rather than quantitative description for the jammer decision process, which has little influence on the optimization result.

4) How the path is decided?

The path is decided by the radar actions and the jammer actions. The radar decides its actions by the proposed optimization method, which conducts the radar winning probability statistics based on the prediction of left game interval. The jammer decides its actions by softmax when facing a radar action. Through the above process, a path is constructed until the left game interval turns negative.

5) On what basis the time interval is decided and optimized?

There are three time intervals proposed in the paper: left game interval, preparation interval and recognition interval of radar and jammer.

The left game interval is decided by the prior knowledge which can be easily got by the radar such as the distance between the radar and target, the velocity of radar and the trajectory. Then it is updated by the confrontation interval of the round. There is no need for optimization.

The preparation interval / recognition interval of radar and jammer is a parameter that is decided by the prior knowledge in the simulation. There is no need for optimization.

Round 2

Reviewer 2 Report

After conducting a comprehensive review of both the original manuscript submitted to Remote Sensing and its subsequent revised version, I am delighted to highlight the remarkable enhancements that have been made. The modifications incorporated in the new iteration have undeniably captivated my attention and garnered my resounding approval. Thus, I am pleased to announce my acceptance of the revised manuscript and express my utmost satisfaction with the revisions implemented.

English Is OK!